# Settlement Characteristic of Warm Permafrost Embankment with Two-Phase Closed Thermosyphons in Daxing'anling Mountains Region

**Guanfu Wang** [1,2], **Jiajun Bi** [2], **Youkai Fan** [2], **Long Zhu** [3], **Feng Zhang** [1,2,*] and **Decheng Feng** [2]

1    CCCC First Highway Consultants Co., Ltd., Xi'an 710075, China
2    School of Transportation Science and Engineering, Harbin Institute of Technology, Harbin 150090, China
3    Guizhou Police College, Guiyang 550005, China
\*    Correspondence: zhangf@hit.edu.cn

**Abstract:** The Xing'anling Mountains are the second largest permafrost region in China. One of the important issues for highways in these regions is how to control the settlement during the operation period to meet the demand of road stability. This paper selects a typical permafrost embankment in the Daxing'anling Mountains permafrost region, presents the finite element models of the embankment, and verifies it using field monitoring data to study the thermal and deformation characteristics within 50 years after construction. Calculation results illustrate that the permafrost under the embankment has degraded significantly during the operation period of the highway and led to serious settlement. To prevent the degradation of permafrost, a series of models with two-phase closed thermosyphons (TPCTs) were established to analyze the cooling effect. The contribution of different factors, including install locations, depth, and shapes of the TPCTs, were assessed on their effects on cooling the permafrost and reducing the embankment settlement. Results show that the TPCTs have an excellent cooling effect on the permafrost embankment. However, as the TPCTs change the temperature distribution of the embankment, they will inevitably cause differential settlement. In order to ensure the cooling effect and reduce the differential settlement of the embankment, it is suggested that L-shaped TPCTs should be adopted in the remedial engineering.

**Keywords:** permafrost region; embankment; on-site monitoring; finite element method; thermal characteristics; two-phase closed thermosyphon



## 1. Introduction

Permafrost accounts for 25% of the total land area of the world, and 22.4% of the land area in China [1]. Countries around the world have carried out a large number of engineering activities in permafrost regions, and the main problem faced by engineering construction in permafrost regions is thaw settlement by permafrost degradation [2,3]. This problem is more obvious in the context of global climate change, which seriously affects the safe operation of construction in cold regions [4–7].

Permafrost in northeast of China covers about $3.1 \times 10^5$ km$^2$ [8], and a large number of asphalt pavement and fewer concrete pavement highways were constructed on it, such as the Beijing–Mohe highway (G111), Dandong–Altay highway (G331), and Jiayin–Linjiang highway (G222), as shown in Figure 1. Researchers have shown that the global permafrost temperature increased by $0.29 \pm 0.12$ °C between 2007 to 2016 [9], and the permafrost in northeast of China has increased by 0.2–0.5 °C during the past six years [10]. It is obvious that the permafrost in Northeast of China is dying, and the aera is shrinking and the north boundary of Eurasian permafrost is moving at a rate of $3.6 \times 10^4$ km$^2$ per decade [8]. This results in road diseases, such as wavy surface, thaw settlement, and longitudinal cracking of highways. The situation is similar with the Qinghai–Tibet Highway, which is located in permafrost, although the Daxing'anling Mountains are characterized by

lower altitudes, colder temperatures, longer duration of winter, and hotter temperatures and longer sunshine periods in summer compared to the Qinghai–Tibet high-altitude permafrost regions [11,12]. These road problems can mainly be attributed to the changes of thermal distribution, which cause thaw consolidation and creeping of the permafrost foundation [13,14].

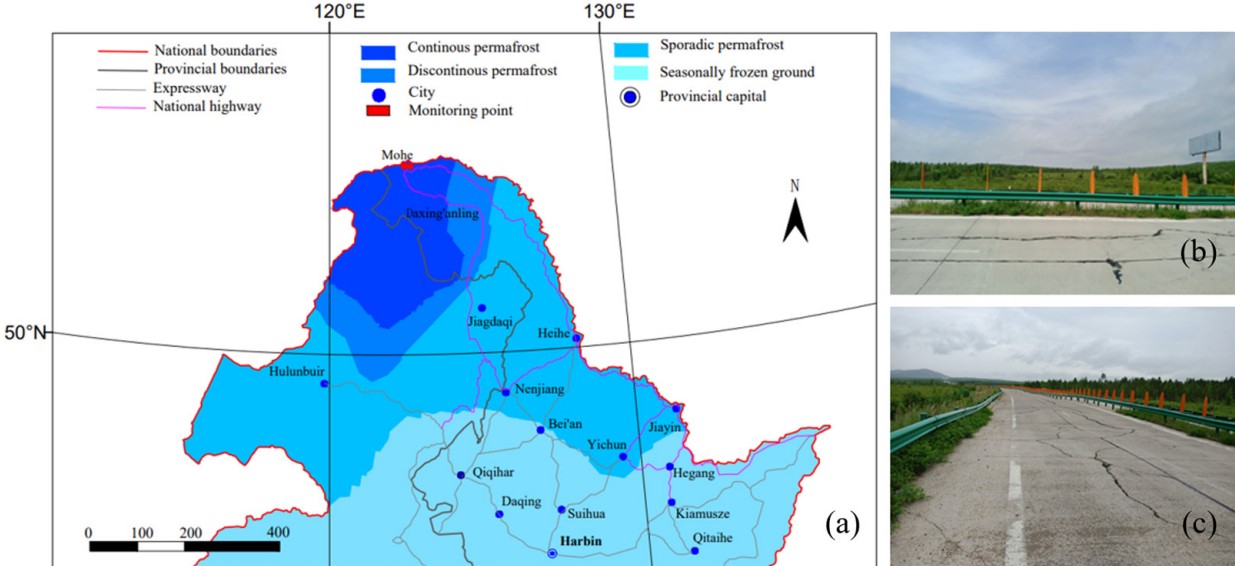

**Figure 1.** Permafrost distribution in Northeast China and the problems of the Mohe–Beijicun highway: (**a**) permafrost distribution in Northeast China; (**b**) Mohe–Beijicun Highway K31 + 700; (**c**) Mohe–Beijicun Highway K31 + 765.

Two-phase closed thermosyphon (TPCT) uses the principle of thermosyphon to drive the working medium in the TPCT (e.g., freon, ammonia, propane) to continuously generate vapor–liquid two-phase convective circulation. It is composed of an evaporator section, an adiabatic section, and a condenser section. The evaporator section is buried in the soil and is in close contact with the soil, which is the main cooling section. The liquid working medium is mainly vaporized in the evaporator section to bring out the heat. The condenser section is the top part of the TPCT. It is the main heat radiating section, which is directly exposed to the air and takes away the heat via air convection [15,16]. TPCT is one effective method for cooling the foundation and is generally used in new construction of permafrost highway and railway engineering projects [17,18]). Differently from railways, highways have a wider embankment, and will encounter more difficulties in using TPCTs to cool down the permafrost. Recently, researches on the temperature distribution characteristic of highway TPCT embankments have mainly been based on finite element calculation. Forsstrom et al. [19] studied a TPCT test section of Chena Hot Springs Road in Alaska and found that the TPCTs could effectively increase freezing depth. Markov et al. [20] improved the calculation method of freezing radius around two-phase thermosyphons in areas with a subarctic climate. Yu et al. [21] studied the crack formation of two sections of the Qinghai–Tibet highway embankment installed with TPCTs using field monitoring and temperature numerical simulation results. Pei et al. [22] simulated the geotemperature control performance of TPCTs in the shady and sunny slopes of an embankment of the Qihai–Tibet Highway. Pei et al. [23,24] established a series of highway embankment geotemperature models, and evaluated the cooling effect of L-shaped TPCTs and the influence of the installation position on the spatial heat control. Kukkapalli et al. [25] developed some roadway embankment models with differently shaped TPCTs to seek an optimal spacing between TPCTs.

Thaw deformation of the road permafrost foundation is a multiphysics process. Numerous studies have been carried out to study the deformation characteristic of em-

bankments in permafrost regions. Zhang et al. [26] presented a model that allowed the migration of unfrozen water so as to compute the settlement of warm permafrost. Zhang et al. [27] developed a numerical model to describe the hydro-thermo-mechanical process of a permafrost embankment and analyze the influence of theshady–sunny slope effect. Zheng et al. [28] constructed a thermo-mechanical model including the creep behavior of the warm permafrost layer to predict the settlement of this embankment. Ming et al. [29] established a 2D numerical model that considers the soil compression, thaw consolidation, and creep to analyze the embankment deformation in permafrost regions. Qi et al. [30] used large strain thaw consolidation theory to study the thaw consolidation behaviors of permafrost roadway embankments. Although there have been some research results on the settlement of permafrost embankments, studies of the calculation of the deformation characteristics of TPCT embankments are still few [31,32], and there is a lack of comprehensive calculations on the frost heave, thaw settlement, and creep of warm permafrost of TPCT embankments.

However, while the current research on TPCT embankments has focused on the construction stage, investigation on using TPCTs to reduce and control thaw-settlement deformation to satisfy the pavement requirements in warm and ice-rich permafrost for highway maintenance is still lacking. Therefore, this research aimed to investigate the long-term thermal and deformation characteristics of highway embankments in the Daxing'anling Mountains permafrost region, and to explore the cooling effect of different types of TPCTs and the improvement effect on the differential settlement of pavement. A typical section of embankment along the Mohe–Beijicun Highway was selected for temperature and deformation monitoring, and then a thermal–mechanical finite element model of TPCT permafrost embankments was proposed and verified with measured data. Finally, the influences of TPCTs, including locations, depth, and shapes, were analyzed and discussed based on the thermal and deformation performances during a 50 year road operation period.

## 2. Coupled Thermal–Mechanical Modeling

### 2.1. Coupled Thermal–Mechanical Model

2.1.1. Physics Models

Heat transfer in a permafrost embankment can be approximately regarded as a two-dimensional solid heat conduction problem. The heat conduction equation can be expressed as Equation (1):

$$\rho L \frac{\partial f_S}{\partial t} + \nabla \cdot (\lambda \cdot \nabla T) - \rho C \frac{\partial T}{\partial t} = 0 \tag{1}$$

where $T$ is the temperature (°C), $t$ is time (s), $\lambda$ is the thermal conductivity (W/(m·°C)), $C$ is the specific heat capacity (J/(kg·°C)), $\rho$ is the density of the material (kg/m$^3$), $\nabla$ is the Hamiltonian differential operator, and $L$ is the latent heat (J/kg); $f_s$ is the solid fraction, which indicates the proportion of ice in the two phases of ice and water, which can reflect the degree of phase change of frozen soil, and its expression is Equation (2):

$$f_S = (T_L - T)/(T_L - T_S) \tag{2}$$

where $T_L$ is the upper limit of the phase change region and $T_S$ is the lower limit of the phase change region. When the soil temperature reaches $T_L$, $f_s$ is 0, which indicates that the soil has completely thawed; when the soil temperature reaches $T_S$, $f_s$ is 1, which indicates that the soil has completely frozen.

In the phase change region, the thermal parameter, $\lambda$, is no longer a thermal parameter in a single state, but becomes the temperature-related functions $\lambda$ and $C'$ shown in Equations (3) and (4) [33]:

$$\lambda = \begin{cases} \lambda_f & T \leq T_f \\ \lambda_f + \frac{\lambda_u - \lambda_f}{\Delta T}[T - T_S] & T_f < T < T_u \\ \lambda_u & T \geq T_u \end{cases} \tag{3}$$

$$C' = \begin{cases} C_f & T \leq T_f \\ C_f + \frac{C_u - C_f}{\Delta T}[T - T_S] + \frac{L}{\Delta T} & T_f < T < T_u \\ C_u & T \geq T_u \end{cases} \tag{4}$$

where $C'$ is the equivalent specific heat capacity after considering the phase change, $\lambda_u$ and $\lambda_f$ are the thermal conductivity of the soil in the thawing and freezing state, $C_u$ and $C_f$ are the specific heat capacity in the thawing and freezing state, and $\Delta T$ is the length of phase change region.

Replace Equation (1) with Equation (5):

$$\left(\rho L \frac{\partial f_S}{\partial T} - \rho C\right)\frac{\partial T}{\partial t} + \nabla \cdot (\lambda \cdot \nabla T) = -\rho C' \frac{\partial T}{\partial t} + \nabla \cdot (\lambda \cdot \nabla T) = 0 \tag{5}$$

For the plane strain problem, the stress–strain constitutive relation of the soil under load can be given by Equation (6):

$$\{\sigma\} = [D]\left(\{\varepsilon\} - \{\varepsilon^t\}\right) \tag{6}$$

where $\{\sigma\}$ is the total stress, $\{\sigma\} = \{\sigma_x\ \sigma_y\ \tau_{xy}\}^T$, $\{\varepsilon\}$ is the total strain, $\{\varepsilon\} = \{\varepsilon_x\ \varepsilon_y\ \gamma_{xy}\}^T$, $\{\varepsilon^t\}$ is the temperature strain, $\{\varepsilon^t\} = \{\varepsilon^t_x\ \varepsilon^t_y\ \gamma^t_{xy}\}^T$, and $[D]$ is the elastic matrix shown in Equation (7):

$$[D] = \frac{E(1-\mu)}{(1+\mu)(1-2\mu)} \begin{bmatrix} 1 & \mu/(1-\mu) & 0 \\ \mu/(1-\mu) & 1 & 0 \\ 0 & 0 & (1-2\mu)/2(1-\mu) \end{bmatrix} \tag{7}$$

The settlement of the permafrost layer can be calculated by the thawing–settling coefficient $\delta_0$, which comprehensively reflects the warm creep and thawing shrinkage of permafrost during the process of thawing.

In this study, $-1.5\,°C$ was taken as the dividing line between warm permafrost and cold permafrost. The warm permafrost region produces 40% temperature strain and 60% of the temperature strain is generated in the phase change region. Therefore, the vertical temperature strain of thaw settlement can be obtained as Equation (8):

$$\varepsilon^t_v = -\delta_0 \cdot \begin{cases} 0 & T \leq T_C \\ \frac{0.4}{T_S - T_C} \cdot (T - T_C) & T_C < T < T_S \\ \frac{0.6}{T_L - T_S} \cdot (T - T_S) + 0.4 & T_S < T < T_L \\ 1 & T \geq T_L \end{cases} \tag{8}$$

where $-\delta_0$ indicates that the temperature strain is reduced by compression, and $T_C$ is the dividing line between warm permafrost and cold permafrost, for which in this study $-1.5\,°C$ is used [13].

When the thawing shrinkage deformation in the horizontal direction is not considered, the temperature strain components in each direction are as in Equation (9):

$$\varepsilon^t_y = (1+\mu) \cdot \varepsilon^t_v\,,\ \varepsilon^t_x = 0\,,\ \gamma^t_{xy} = 0 \tag{9}$$

where $\varepsilon^t_x$ and $\varepsilon^t_y$ are the normal strain caused by the decrease of the volume caused by frozen soil thawing. $\gamma^t_{xy}$ is the shear strain caused by the decrease of volume caused by frozen soil thawing.

Substituting Equation (9) into Equation (8), and shown as Equation (10):

$$\varepsilon^t_y = -(1+\mu)\delta_0(T - T_C) \cdot \begin{cases} 0 & T \leq T_C \\ \frac{0.4}{T_S - T_C} & T_C < T < T_S \\ \frac{0.6}{T_L - T_S} \cdot \frac{T - T_S}{T - T_C} + \frac{0.4}{T - T_C} & T_S < T < T_L \\ \frac{1}{T - T_C} & T \geq T_L \end{cases} \tag{10}$$

Introducing the linear thermal expansion coefficient, making $\varepsilon^t_y := \cdot \alpha \cdot (T - T_C)$, then $\alpha$ can be expressed as Equation (11):

$$\alpha = -(1+\mu)\delta_0 \cdot \begin{cases} 0 & T \le T_C \\ \frac{0.4}{T_S - T_C} & T_C < T < T_S \\ \frac{0.6}{T_L - T_S} \cdot \frac{T - T_S}{T - T_C} + \frac{0.4}{T - T_C} & T_S < T < T_L \\ \frac{1}{T - T_C} & T \ge T_L \end{cases} \tag{11}$$

### 2.1.2. Geometric Model

Figure 2 displays the geological conditions of each layer and the embankment geometric model. Since the highway runs north to south, the shady–sunny slope effect is ignored, and half of the embankment is studied. The road has a 0.25 m thick cement concrete slab, followed by a 0.35 m thick cement treated base (CTB), a 2.3 m height embankment with a ratio of 1:1.5 (V:H), and a 1.1 m height berm on the side of the embankment. The gravelly embankment fill was underlain by a 7 m thick clay layer, followed by a 15 m weathered sandstone layer.

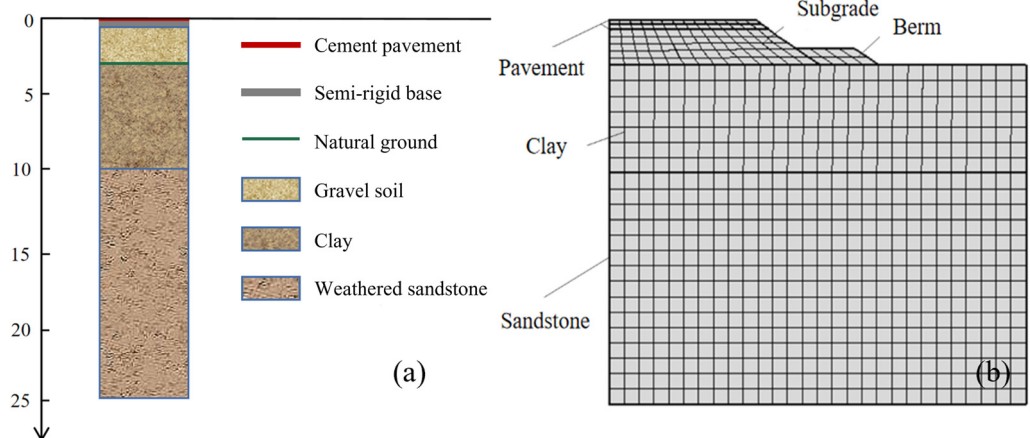

**Figure 2.** Geological conditions and geometric model: (**a**) diagram of geological conditions; (**b**) geometric model and mesh division.

### 2.1.3. Boundary Conditions

Combining Mohe meteorological and field monitoring data, Equations (12)–(14) were selected as the temperature of the upper boundary of pavement surface, berm and slope surface, natural ground surface, and slope surface, respectively. The lower boundary of the model is a natural bedrock with a depth of more than 20 m, so a constant flux boundary of 0.03 W/m² was adopted. Under the boundary condition of Equation (14), when calculating the numerical model before the highway was built, the stable initial temperature field of the permafrost foundation can be obtained.

$$T = 1.2 + 18.2 \sin\left(\frac{2\pi}{365} \cdot t - 1.62085\right) + \beta t \tag{12}$$

$$T = 0.8 + 18.2 \sin\left(\frac{2\pi}{365} \cdot t - 1.62085\right) + \beta t \tag{13}$$

$$T = -1 + 14.9 \sin\left(\frac{2\pi}{365} \cdot t - 1.62085\right) + \beta t \tag{14}$$

where $t$ is the time (day), $\beta$ is the coefficient when the annual average temperature rises are considered and is taken as 0.033 °C/year [9].

### 2.1.4. Material Parameters

According to filed survey data and related documents [34–38], The thermal and mechanical parameters of each soil layer of the embankment and natural foundation are shown in Tables 1 and 2.

**Table 1.** Thermal parameters of each layer.

| Layer | Heat Conductivity Coefficient (W·m$^{-1}$·°C$^{-1}$) | | Specific Heat (J·kg$^{-1}$·°C$^{-1}$) | | Density (kg/m$^3$) | Moisture Content (%) |
|---|---|---|---|---|---|---|
| | $\lambda_u$ | $\lambda_f$ | $C_u$ | $C_f$ | $\rho$ | $\omega$ |
| Concrete pavement | 1.68 | 1.68 | 900 | 900 | 2450 | — |
| Base course | 1.57 | 1.57 | 1000 | 1000 | 2350 | — |
| Embankment | 1.82 | 1.98 | 1463 | 1129 | 1800 | 15 |
| Clay | 1.24 | 1.89 | 2090 | 1588 | 1500 | 30 |
| Sandstone | 1.51 | 1.69 | 877 | 771 | 2000 | 2 |

Note: sub-symbol of u presents unfrozen, and sub-symbol of f presents frozen.

**Table 2.** Mechanical parameters of each layer.

| Layer | $a_1$ | $b_1$ | $a_2$ | $b_2$ | $\rho$ (kg/m$^3$) | $\delta_0$ (%) |
|---|---|---|---|---|---|---|
| Concrete pavement | 31,000 | 0 | 0.15 | 0 | 2450 | — |
| Base course | 2000 | 0 | 0.2 | 0 | 2350 | — |
| Embankment | 34 | 8.37 | 0.41 | 0.052 | 1800 | 3 |
| Clay | 5 | 38.84 | 0.35 | 0.036 | 1500 | 25 |
| Sandstone | 140 | 107.81 | 0.25 | 0.004 | 2000 | 1 |

The relation of the elastic modulus $E$ (MPa) and Poisson's ratio $\mu$ with temperature are expressed as Equations (15) and (16):

$$E = \begin{cases} a_1 + b_1(T_m - T)^n & T < T_m \\ a_1 & T \geq T_m \end{cases} \tag{15}$$

$$\mu = \begin{cases} a_2 - b_2(T_m - T) & T < T_m \\ a_2 & T \geq T_m \end{cases} \tag{16}$$

where $T_m$ is the midpoint of the soil phase change region, $a_1$, $b_1$, $a_2$, $b_2$, and $n$ are fitting parameters, and n is generally 0.6 [39,40].

According to the soil quality and total moisture content of the Mohe–Beijicun Highway, the larger value within the range of the thawing settlement coefficient given by the specification is taken according to the most unfavorable situation.

### 2.1.5. Two-Phase Closed Thermosyphon (TPCT)

Assuming that the heat dissipation efficiency of the fins on the TPCT is $\eta = 0.8$, and ignoring the thermal resistance of the TPCT itself and the contact thermal resistance between the evaporator section and the soil, the heat transfer amount Q of the TPCT is:

$$Q = \alpha F(T_a - T) \cdot \eta \tag{17}$$

where $\alpha$ is the convective heat transfer coefficient of the air and the condenser section (W·m$^2$·°C$^{-1}$), $F$ is the effective cooling area of the condenser section (m$^2$), $T_a$ is the ambient temperature outside the embankment (°C), $T$ is the soil temperature around the evaporator section (°C), and $\eta$ is the heat dissipation efficiency of the fins in the condenser section.

The convective heat transfer coefficient $\alpha$ between the air and the condenser section can be calculated by Equation (18) [41]:

$$\alpha = 2.75 + 1.51v^{0.5} \tag{18}$$

where $v$ is the annual average wind speed above the highway (m/s). It is 2.5 m/s according to the meteorological data of Mohe.

The heat flux $q$ applied to the evaporator section is Equation (19):

$$q = \frac{Q}{\pi d_0 l} = \frac{\alpha F \eta}{\pi d_0 l}(T_a - T) \tag{19}$$

where $d_0$ is the outer diameter of the TPCT (m) and $l$ is the length of the evaporator section (m).

According to Fourier's law, the boundary heat source of the TPCTs in the evaporator section satisfies the conditions:

$$\frac{\alpha F \eta}{\pi d_0 l}(T_a - T) = -\lambda \cdot \nabla T \tag{20}$$

### 2.2. Model Verification

#### 2.2.1. On-Site Monitoring

Figure 3 shows the layout of sensors at Section K31 + 700. Four monitoring holes were drilled, and temperature coefficient thermistor temperature sensors with an accuracy of $\pm 0.05$ °C and single point deformation sensors with an accuracy of $\pm 0.05$ mm were installed. Figure 4 shows pictures of the field works. All sensors were collected and connected to the automatic data acquisition box, and a wireless transceiver module was used to transform the monitoring data. The sampling frequencies of the temperature and settlement deformation were 1/2 h and 1/15 day, respectively.

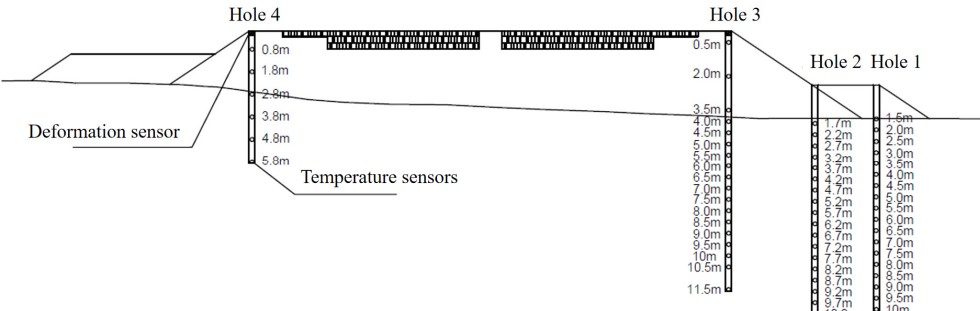

**Figure 3.** The layout of site monitoring devices.

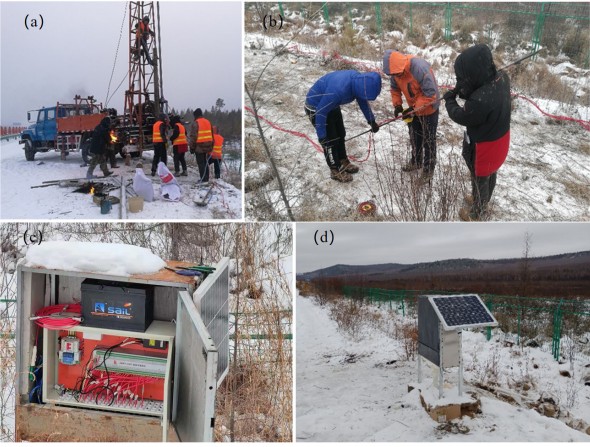

**Figure 4.** Pictures of fieldwork: (**a**) drilling holes; (**b**) fixing position of sensors; (**c**) temperature acquisition module and wireless transceiver module; (**d**) data acquisition box.

#### 2.2.2. Model Validation

Figure 5 compares the calculated ground temperature distribution curve at hole 3 with the on-site monitoring data on 1 January 2019. It illustrates that the numerical calculation results have good spatial consistency with the measured values during the same period. Thus, the calculated and the measured values in hole 3 exhibit good consistency.

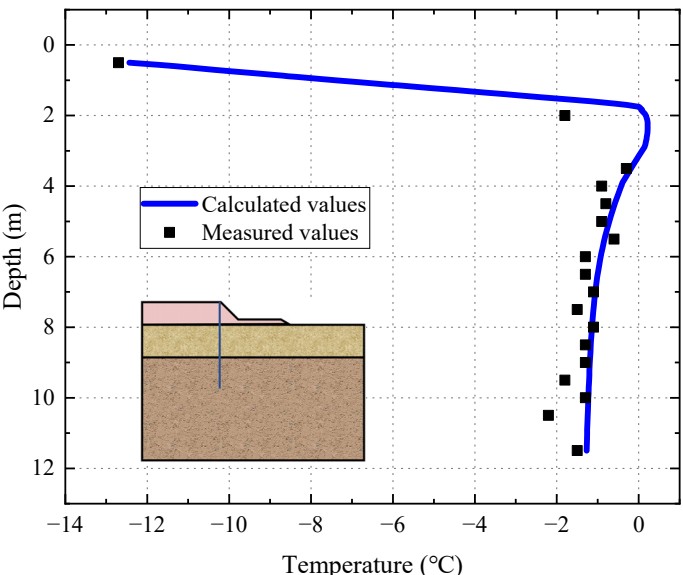

**Figure 5.** Comparison of calculated and measured temperature values of hole 3.

Figure 6 shows the ground temperature curves of measured and calculated values at hole 3. The ground temperature within 3 m gradually decreased from January to February, with a temperature lower than −12 °C, and then gradually rose from March to June. From May, the surface soil gradually thawed into a positive temperature. In November, the temperature was back to negative again. Therefore, The ground temperature curve calculated by the numerical value is in good agreement with the measured value in terms of time evolution.

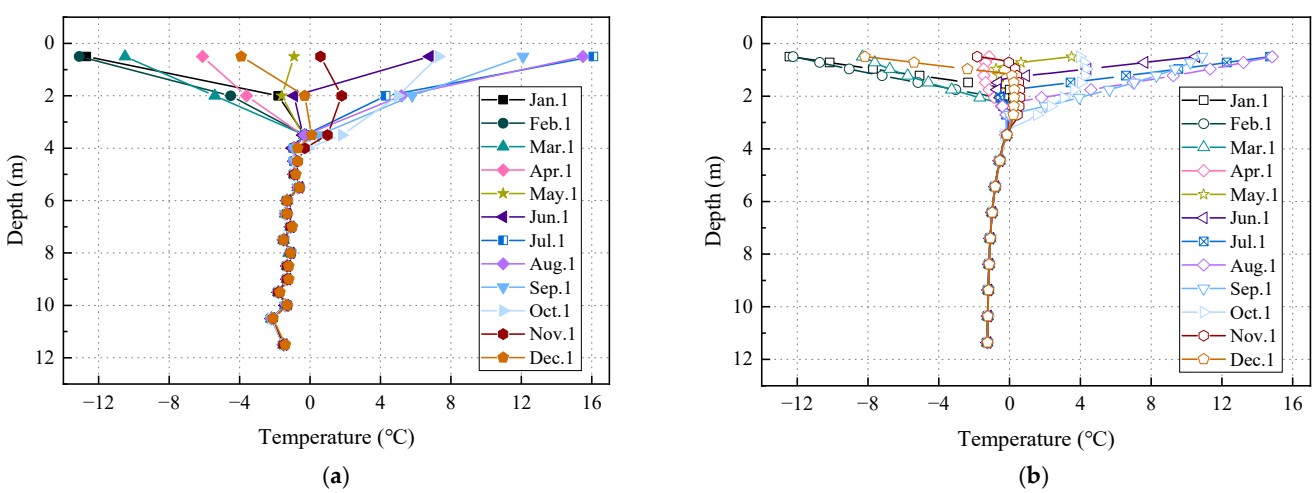

**Figure 6.** Ground temperature curves: (**a**) measured values; (**b**) calculated values.

Figure 7 shows measured and calculated results of embankment settlement at hole 3. The cumulative settlement started from March and cumulative settlement was calculated for three years. Because the larger values were selected in the Equations (16) and (17) to reflect the most unfavorable situation, the calculated settlement value is slightly larger

than the measured value. It can be found that the development trend of the calculated and measured cumulative settlements is closer, and the annual settlement increased by about 35 mm/year. Therefore, this model has certain rationality and reliability for the calculation of the temperature and deformation field, and the calculation results can truly reflect the actual embankment conditions.

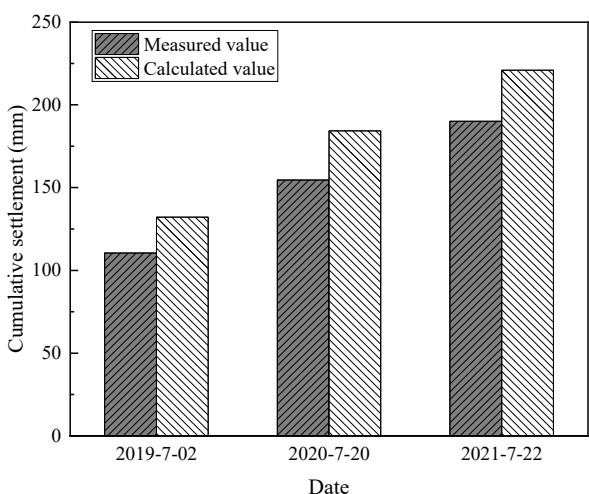

**Figure 7.** Comparison of calculated and measured settlement values of hole 3.

## 3. Evolution Characteristics without TPCT

### 3.1. Temperature Field in 50 Years

Figure 8 shows the ground temperature at the centerline of the embankment from 2010 to 2060 on October 1. In the early stage after embankment construction, the temperature of the embankment is significantly higher than that of the natural foundation. Due to the influence of the original frozen soil in the natural foundation under it, the temperature of the embankment decreases slightly after 10 years of embankment construction in 2020. At the same time, affected by the thermal disturbance of the embankment, the temperature of the foundation under it has increased remarkably. During the 40 years from 2020 to 2060, the temperature of embankment and foundation gradually increases, and the increased range of the embankment is significantly greater than that of the foundation due to the increase of the annual average temperature outside. The permafrost under the embankment has been seriously degraded, the permafrost table has decreased rapidly, and the permafrost temperature is in a severe phase change region.

Figure 9 shows the variation of ground temperature at different layers over 50 years. We selected different depths of embankment and foundation for analysis. For the centerline of the embankment, the temperature of the foundation rises rapidly within five years after the construction of the embankment. Then the temperature fields of the embankment and natural foundation are balanced with each other and begin to show a linear warming trend. At this time, the influence of the external temperature begins to appear. The average annual warming rate of the soil at the top, middle, and bottom of the embankment is 0.092, 0.101, and 0.11 °C/year, respectively. The temperature rising rate of permafrost is faster with the shallow depth. The average annual warming rate of the frozen soil at 3, 5, and 10 m below the ground surface is 0.028, 0.019, and 0.016 °C/year, respectively. In addition, after the permafrost in the foundation thaws, its warming rate is accelerated. Among them, the annual average heating rate of the thawing soil at 0.5, 1, 1.5, and 3 m below the ground surface is 0.095, 0.085, 0.08, and 0.039 °C/year, respectively.

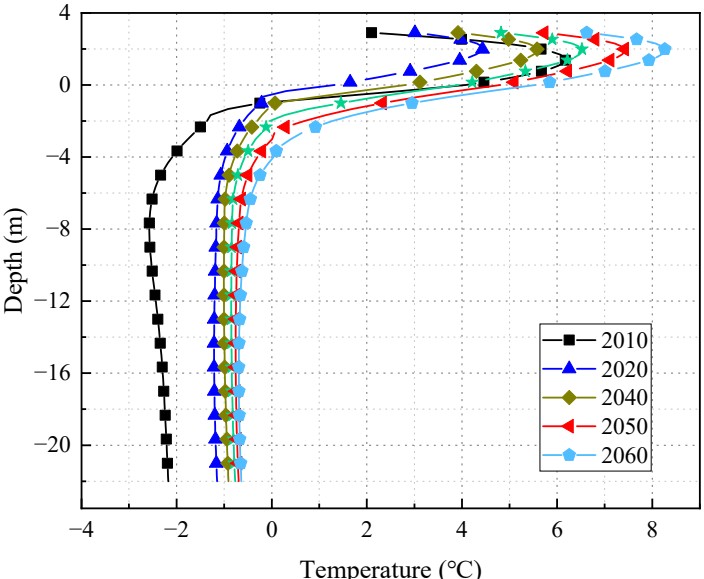

**Figure 8.** Ground temperature curves of embankment centerline from 2010 to 2060.

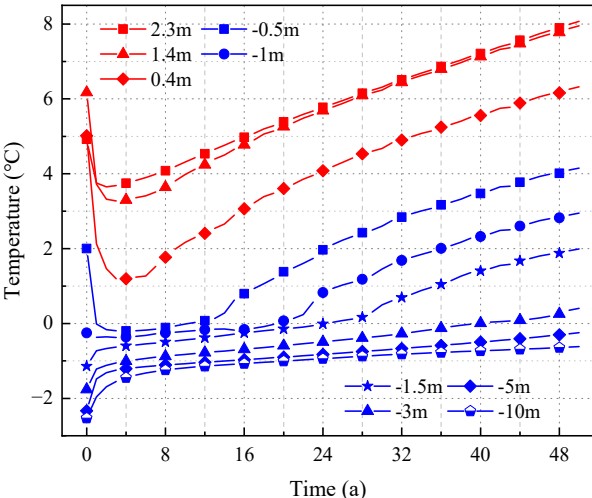

**Figure 9.** Variation curves of embankment centerline temperature in different layers with time.

### 3.2. Deformation Field in 50 Years

Figure 10 shows the settlement evolution of the embankment at the maximum thawing depth (October 1) from 2010 to 2060; the x and y axes are the horizontal distance from the centerline of the road and the vertical distance from then nature ground surface, respectively. The settlement increased rapidly in the first 10 years (from 2010 to 2020) after the road construction, and the maximum settlement increased from 0 cm to 82.7 cm. In the next 40 years, the soil settlement growth rate slows down, and the maximum settlement increases by 29.3, 26, 23, and 20 cm every 10 years. In addition, according to the density of isoline, the settlement in the embankment section is mainly concentrated in the embankment range, and with time, the settlement range gradually expands to one side of the ground surface. The settlement of the soil layer under 7 m under the surface is uniform, while the settlement within 7 m under the ground surface is uneven. The settlement under the embankment is more obvious than that under one side of the ground surface at the same depth.

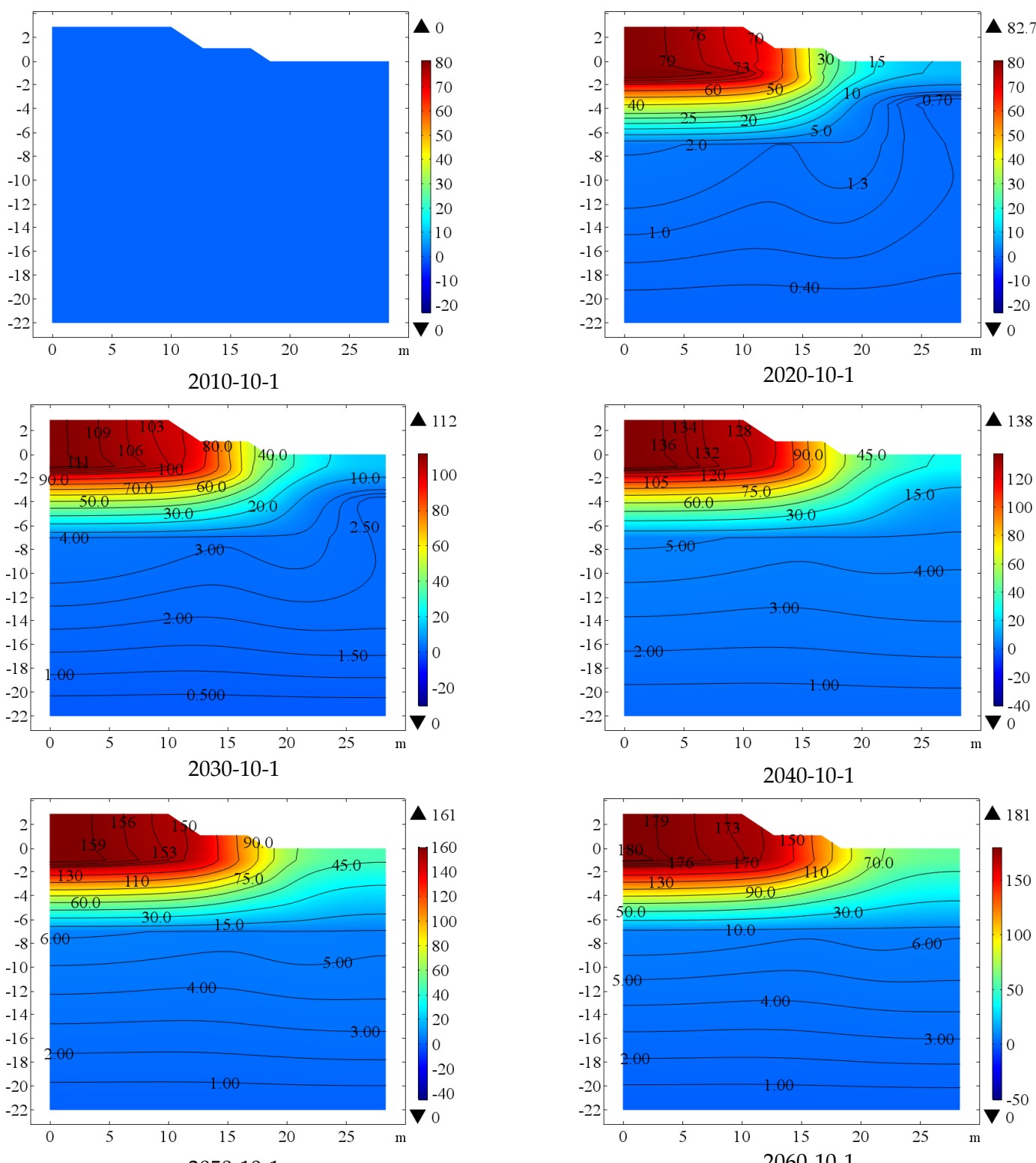

**Figure 10.** Embankment settlement from 2010 to 2060 (cm).

Figure 11 shows the settlement at the centerline of the road surface, the shoulder, the centerline of the berm, and the natural ground surface during 50 years after the embankment construction. The settlement rate of the embankment gradually slows down with time. In the first five years, the embankment settlement develops rapidly. The centerline of the embankment has a total settlement of 60 cm, and the annual average settlement rate is

12 cm/year. The total settlement of the shoulder and berm is 51 and 29 cm, and the average annual settlement rate is 10.2 and 5.8 cm/year, respectively. In the following 45 years, the embankment settlement develops linearly, and the settlement rates at the centerline of the embankment, the shoulder, and the berm surface are relatively consistent at about 2.7 cm/year. However, the settlement development rate on one side of the surface gradually becomes faster with time. In the first five years, the settlement rate is about 0.9 cm/year, and in the last five years, the rate increases remarkably and is about 1.5 cm/year. Therefore, calculation results display that with the continuous warming of warm permafrost, the settlement of the embankment without remedial measures will continue to develop.

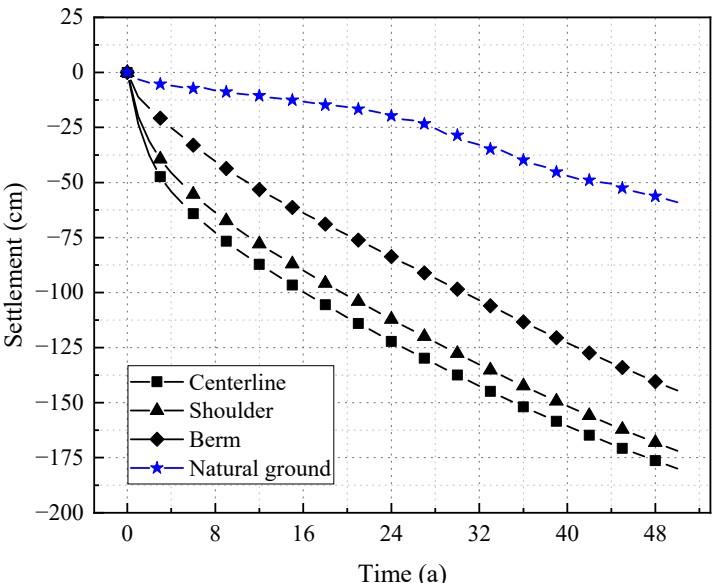

**Figure 11.** Change of settlement in different locations.

## 4. Influence of TPCTs

### 4.1. Locations of TPCTs

Figure 12a shows the ground temperature 15 m below the ground surface at the centerline of the embankment when the TPCTs are arranged at the road shoulder, slope toe, and berm. The length of the evaporator section is 5 m, the adiabatic section is 1.9 m, and the condenser section is 1.5 m. The fins of the condenser section are 3 m$^2$, and the outer diameter of the TPCTs is 0.1 m. With the increase of using time, the ground temperature shows a trend of first a decline and then a rise. The closer the TPCTs are arranged to the road shoulder, the faster the cooling rate at the centerline of the embankment before reaching the minimum temperature. Meanwhile, it indicates that the closer it is to the TPCTs, the more obvious the cooling effect of the soil, and the stronger the ability to resist the influence of external heating. When the TPCTs are installed on the shoulder, the toe of the slope, and the outer shoulder of the berm of the embankment, the annual average cooling rate of ground temperature is 0.107, 0.066, and 0.037 °C/year, respectively.

Figure 12b shows the ground temperature distribution at the centerline of the embankment after 10 years of operation of the three cases. The horizontal locations of the TPCTs have almost no effect on the soil temperature inside the embankment and the soil temperature in the shallow layers of the foundation. The impact range is mainly in the deep layers below the permafrost table. The TPCTs at the road shoulder has the best cooling effect, and the closer the location is to the road shoulder, the lower the temperature of the permafrost layers. When TPCTs are installed at the shoulder, the toe of the slope, and the berm of the road, the temperature of permafrost is −2.5, −2.1, and −1.8 °C, respectively.



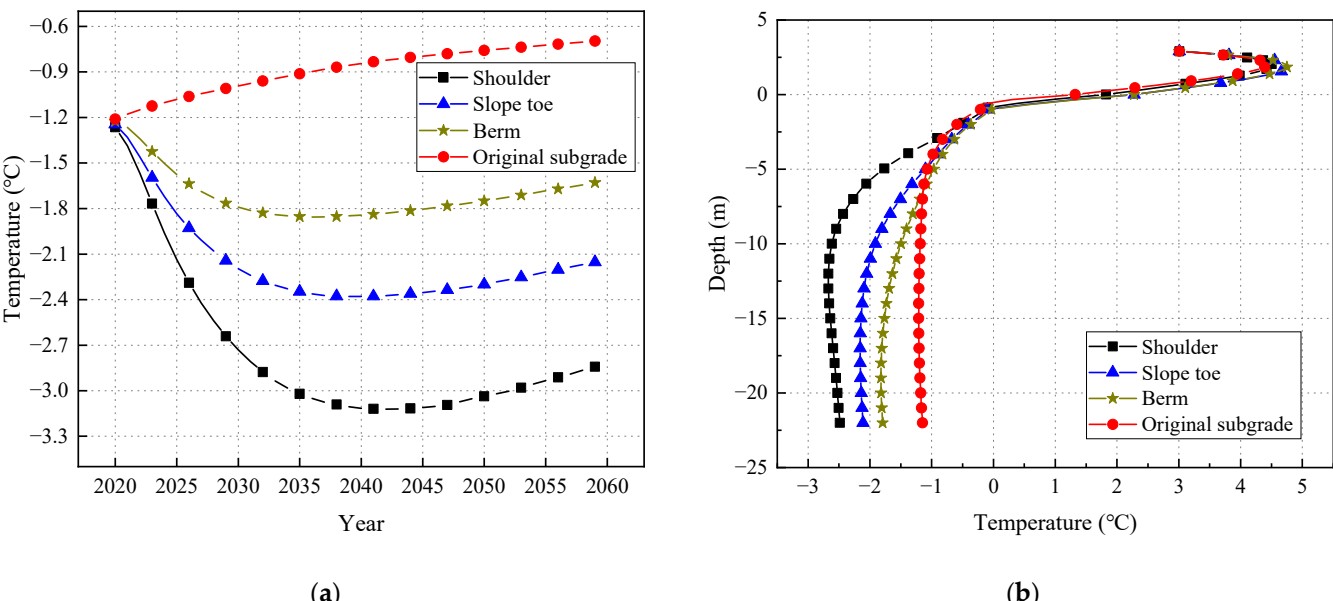

(**a**)                                                                        (**b**)

**Figure 12.** The temperature change of the embankment centerline with different locations of TPCTs: (**a**) time-history curve of embankment temperature; (**b**) temperature of embankment after 10 years.

Figure 13 shows the distribution of the surface settlement when the TPCTs are placed in three different locations after one year. The different lateral location of the TPCTs have significantly improved the surface settlement within the range of the road surface. It can be found that changing the lateral location of the TPCTs significantly changes the lateral difference of ground settlement. When the location of the TPCTs is the road shoulder, the improvement effect of the settlement within the road area is much more obvious, but it also causes serious differential settlements; the maximum differential settlement value reached 60.6 cm.

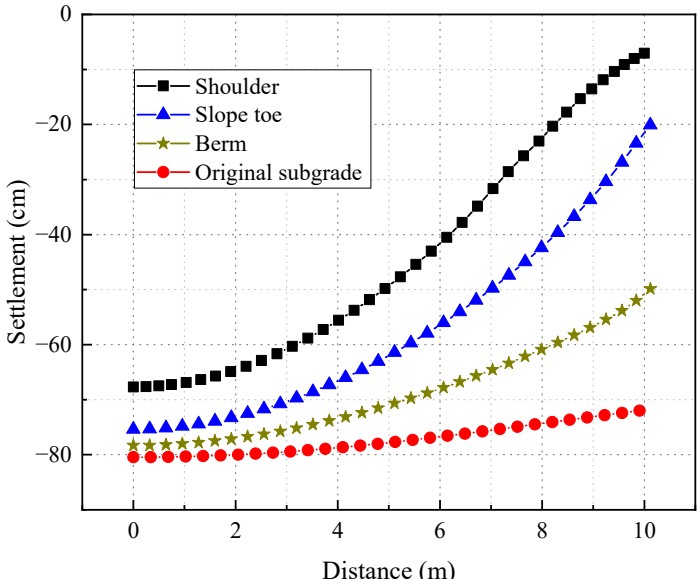

**Figure 13.** Lateral distribution of road surface settlement after 1 year with different locations of TPCTs.

### 4.2. Depth of TPCTs

Figure 14a indicates the ground temperature 15 m below the ground surface at the centerline of the embankment when the TPCTs are buried at three different depths. The TPCTs were set at the foot of the embankment slope, the evaporator section was 4, 6, and

8 m below the surface, and the length of the evaporator section was 5 m. The ground temperature curves of the TPCT embankment have a similar changing trend. As the running time of the TPCTs increases, the ground temperature also shows a trend of first a decline and then a rise. When the TPCTs are buried deeper, the permafrost can obtain a better cooling effect at the centerline before reaching the minimum temperature. This demonstrates that increasing the buried depth of the evaporator section of a TPCT can effectively reduce the annual average temperature in the deep layer of the permafrost foundation, and it has a stronger ability to resist the influence of external heating during the operation period.

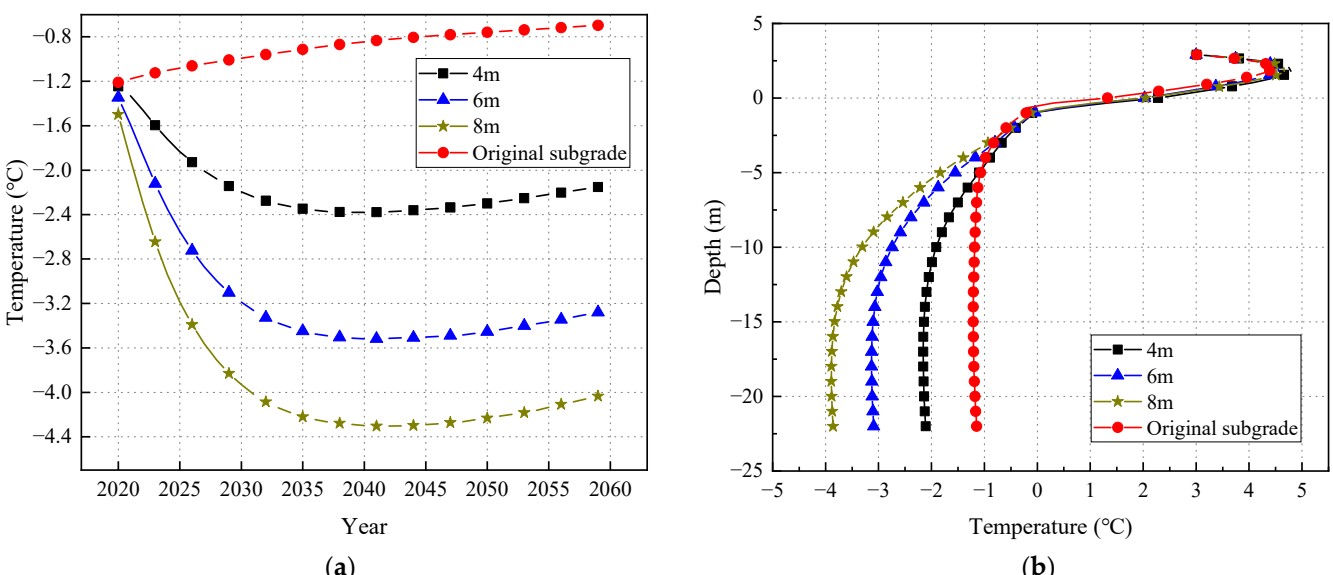

**Figure 14.** The temperature change of the embankment centerline with different depths of TPCTs: (**a**) time-history curve of embankment temperature; (**b**) temperature of embankment after 10 years.

Figure 14b shows the ground temperature distribution at the centerline of the embankment when the TPCTs are buried at three different depths at 10 years. With the increase of burying depth of TPCTs, the temperature of the permafrost decreases more obviously. It can be explained that the soil temperature is distributed from higher to lower along the depth direction affected by the negative external temperature during the annual cycle of the TPCTs' work. When the burying depths of TPCTs are deeper, the temperature difference between the surrounding soil and the outside temperature becomes greater. At this time, it is easier for the TPCTs to reach the starting temperature, their working time is prolonged, and they have more cooling capacity.

Figure 15 shows the distribution of the surface settlement under the embankment when the TPCTs are buried at three different depths one year after the application of the TPCTs. The TPCTs with different depths have significantly improved the surface settlement within the road surface, but the unevenness of the settlement has become more obvious. When the insertion depth of the TPCT is 8 m, the clay layer 3 m from the surface cannot be cooled well, resulting in relatively large settlement around the TPCT, that is, there is a larger settlement value on the right side of the green curve in Figure 15. Meanwhile, according to Equation (17), the cooling capacity of the TPCT is related to the soil temperature around the evaporator section. According to the temperature distribution of the foundation, the deeper the burial depth, the lower the soil temperature and the larger the total cooling capacity. Therefore, after one year's temperature transfer, the soil temperature near the embankment centerline is relatively low, which is also reflected in Figure 14a. Accordingly, there is a large settlement value on the left side of the green curve in Figure 15. When the buried depths of the TPCTs are 4, 6, and 8 m, the differential settlement within the road area is 55.3, 58.6, and 44.4 cm, respectively.

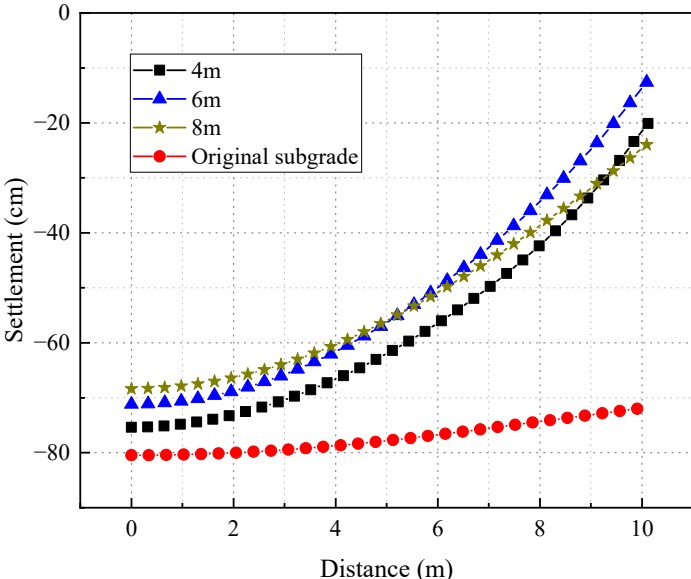

**Figure 15.** Lateral distribution of road surface settlement after 1 year with different depths of TPCTs.

### 4.3. Shapes of TPCTs

The common installation of TPCTs is vertically installed, obliquely installed, and L-shaped. The different arrangements of TPCTs have different inclination angles concerning the vertical direction of the evaporator section of the TPCTs. The traditional L-shaped TPCTs have an inclined lower part and a vertical upper part, which cannot improve TPCTs inserted into the ground farther than the ones obliquely installed to obtain a larger horizontal cooling range. Therefore, a type of flexible L-shaped TPCT was designed to obtain a larger cooling radius, as shown in Figure 16. The evaporator sections are all set as 5 m and buried into the embankment from the slope toe.

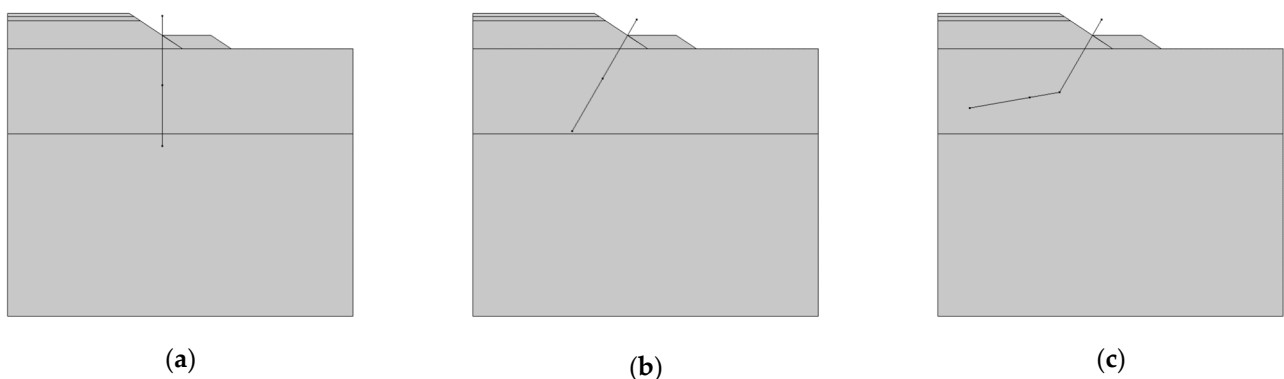

**Figure 16.** Different shapes of TPCTs: (**a**) vertically installed TPCT; (**b**) obliquely installed TPCT; (**c**) flexible L-shaped TPCT.

Since the condensed reflux of the working fluid in the TPCTs needs to rely on the action of gravity, the TPCTs with different inclination angles have different speeds at which the working fluid returns to the evaporator section, resulting in different refrigeration efficiencies in the evaporator section. When simulating different shapes of TPCTs, different heat fluxes are applied to the evaporator section of the TPCTs expressed as Equation (21):

$$q = q_0 \cdot \cos\theta \tag{21}$$

where $q_0$ is the heat flux density of the evaporator section of the vertically installed TPCTs (W/m$^2$), and $\theta$ is the angle between the evaporator section of the TPCTs and the vertical

direction. The angles of vertically installed, obliquely installed, and L-shaped TPCTs are 0, 30, and 80 degrees, respectively.

Figure 17a shows the ground temperature 15 m below the ground surface at the centerline of the embankment with different shapes of TPCTs. Among them, the obliquely installed TPCTs have the most significant cooling effect on the ground temperature at the centerline of the embankment, and its cooling rate is fastest before the ground temperature reaches the minimum temperature. At the same time, the vertically installed and L-shaped TPCTs have a similar cooling effect on the ground temperature at the embankment centerline. When the vertically installed, obliquely installed, or L-shaped TPCTs are used, the annual average cooling rate of ground temperature is 0.140, 0.163, and 0.139 °C/year, respectively.

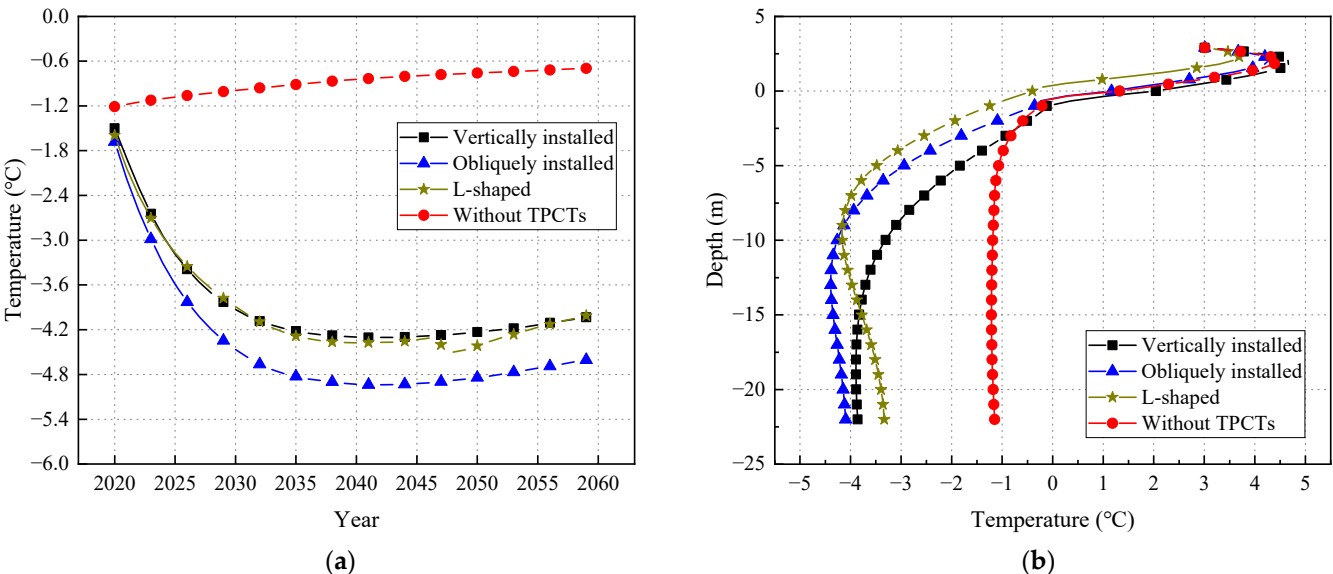

**Figure 17.** The temperature change of the embankment centerline with different shapes of TPCTs: (**a**) time-history curve of embankment temperature; (**b**) temperature of embankment after 10 years.

Figure 17b shows the ground temperature distribution at the centerline of the embankment after 10 years operated using different shapes of TPCTs. The vertically installed and obliquely installed TPCTs have little effect on the soil temperature in the shallow layers. Their influence range is the soil below the permafrost table. For the frozen soil within 2–17 m below the ground surface, the cooling effect of obliquely installed TPCTs is more obvious than those vertically installed, while for the frozen soil below 17 m, there is little difference between them. Compared with the vertically installed and the L-shaped TPCTs, although the L-shaped TPCTs have a lower cooling effect on the temperature of permafrost, they have a wider cooling range in the vertical direction. This not only has a cooling effect on a deep foundation but also has an obvious cooling effect on soil in shallow layers and soil in the embankment.

Figure 18 shows the distribution of the surface settlement under the embankment using different shapes of TPCTs. For the ground surface at the centerline of the embankment, the settlement improvement effect of the L-shaped TPCTs is greater than that of the obliquely installed TPCTs, and the vertically installed TPCTs are the worst. Due to the large inclination angle of the L-shaped TPCTs, their evaporator section can penetrate deep into the foundation, thereby cooling the soil under the centerline of the embankment. Because the cooling range of the vertically installed TPCTs is mainly below the road shoulder, it is unable to produce a good cooling effect on the soil under the centerline of the embankment. Therefore, the L-shaped TPCTs can significantly improve the differential settlement of the road. When the vertically installed TPCTs are used, the differential settlement within the road surface is 44.4 cm; it is 43.1 cm for the obliquely installed TPCTs; and only 6.1 cm using

the L-shaped TPCTs. Therefore, L-shaped TPCTs can improve the differential settlement by more than 86% compared with the other two shapes of TPCTs.

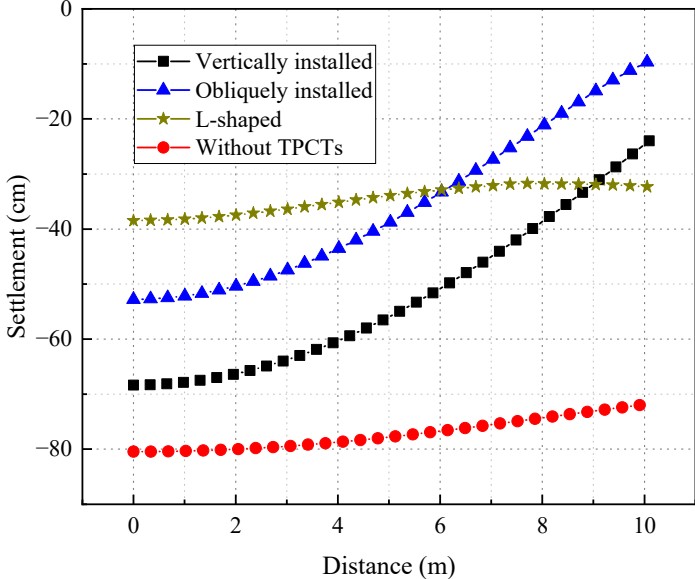

**Figure 18.** Lateral distribution of road surface settlement after 1 year with different shapes of TPCTs.

*4.4. Discussions*

Figure 19 shows the comparison of the temperature at the embankment centerline after 10 years and the differential settlement embankment after 1 year by different cases. Among the influencing factors, the buried depth has the most obvious effect on the cooling effect of the TPCTs. For example, when the burying depth increases from 4 to 6 m, the change in ground temperature after 10 years can reach 1 °C. The cooling effect of TPCTs is comprehensively affected by the cooling capacity and heat transfer distance of the foundation soil, and the cooling capacity is directly related to the temperature (burying depth) of the soil around the TPCT. Hence, the vertically or obliquely installed TPCTs have a better cooling capacity then the L-shaped ones. In addition, the installation of TPCTs at the shoulder has a better cooling effect on permafrost.

The temperature unevenness in the horizontal direction is the main reason for the difference in road surface settlement. When the cooling effect of the TPCTs is more obvious, the effect of improving the settlement is always more remarkable. However, due to the temperature field in the foundation being changed by the TPCTs, there will inevitably be differential settlement. The lateral arrangement locations of the TPCTs and the shapes of the TPCTs have a significant impact on the differential settlement of the embankment. Among them, the L-shaped TPCTs have the best effect on reducing the differential settlement. This is because the cold transfer direction of L-shaped TPCTs is mainly in the vertical direction, which has few disturbances to the temperature uniformity of the soil in the horizontal direction. The L-shaped TPCTs can cause the evaporator section to be close to the center line of the embankment, effectively improving the uneven distribution of the temperature field. Therefore, based on simulation results, it is recommended to install the flexible L-shaped TPCTs to reduce the embankment thawing settlement of the Mohe–Beijicun Highway so as to improve the cooling efficiency of the permafrost and reduce the occurrence of differential settlement.

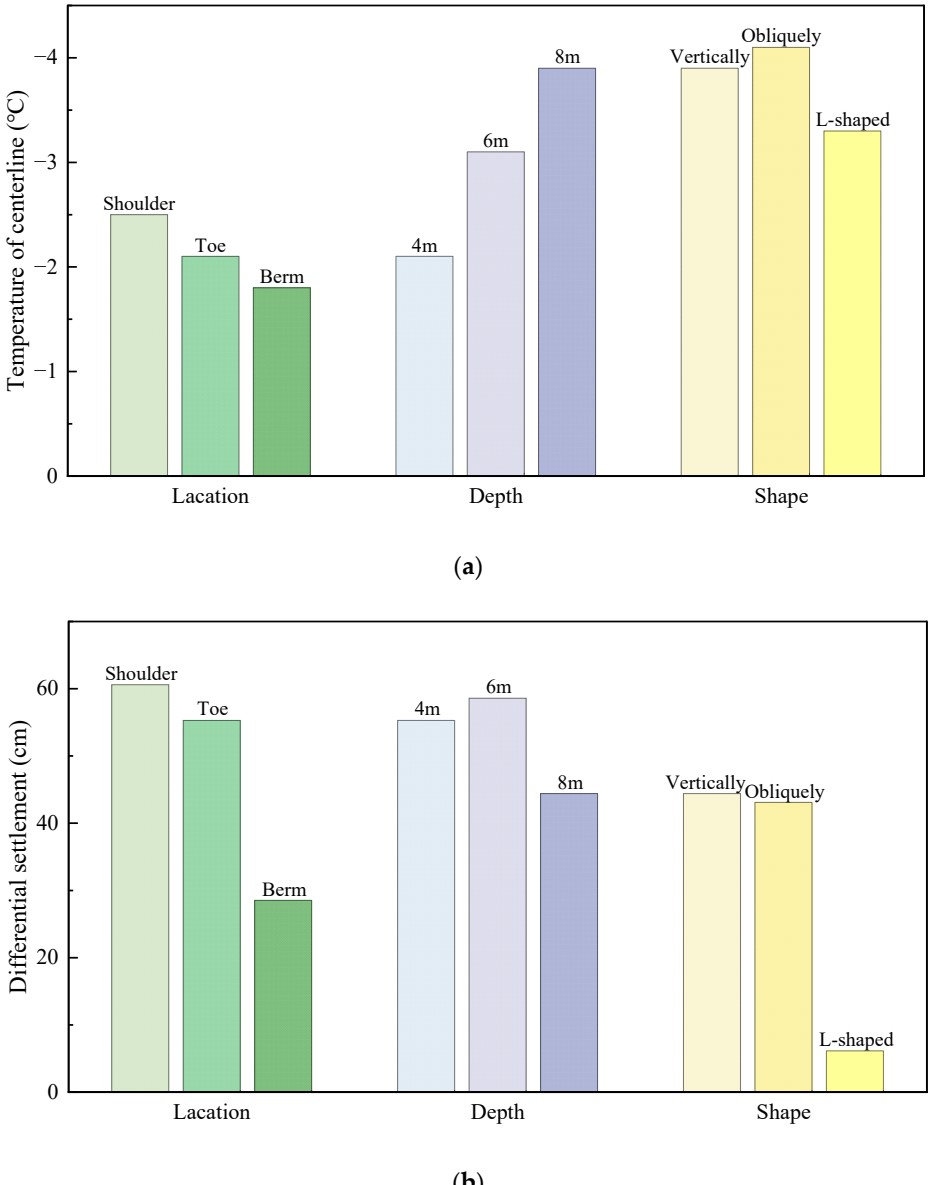

**Figure 19.** Analysis of influencing factors: (**a**) differential settlement embankment after 1 year; (**b**) temperature of embankment centerline after 10 years.

## 5. Conclusions

This study was focused on the long-term temperature and deformation field of the permafrost foundation under the Mohe–Beijicun Highway. Through the finite element method, the changes to the permafrost embankment with and without TPCTs over 50 years were studied. The present study draws the following conclusions:

(1) The permafrost under the embankment of the Mohe–Beijicun Highway will be in the process of warming and thawing in the next 50 years, and the permafrost table has decreased rapidly, which will lead to serious settlement of the embankment. In the first five years, the embankment settlement developed rapidly, and in the following 45 years, the embankment settlement will develop linearly.

(2) TPCTs have an excellent cooling effect on the permafrost embankment. The buried depth of the TPCTs has a great impact on the cooling effect of the soil. When the buried depth is 6 m, the TPCTs has the best cooling effect. TPCTs installed on the shoulder of the embankment can also better cool the permafrost foundation.

(3)  Using TPCTs will inevitably cause differential deformation, especially when the cooling efficiency is better. The locations of the TPCTs and the shape of the TPCTs have a remarkable impact on the differential settlement of the ground within the range of the road surface, and the L-shaped TPCTs have the best effect on improving the differential settlement.

(4)  In order to ensure the cooling effect and reduce the differential settlement of the embankment, it is suggested that installing flexible L-shaped TPCTs should be adopted in the remedial project of the embankment thawing settlement of Mohe–Beijicun Highway.

**Author Contributions:** Conceptualization, F.Z. and D.F.; Data curation, G.W., Y.F. and L.Z.; Software, G.W., J.B. and Y.F.; Validation, F.Z.; Visualization, G.W., J.B. and L.Z.; Writing—original draft, G.W.; Writing—review and editing, J.B., F.Z. and D.F. All authors have read and agreed to the published version of the manuscript.

**Funding:** The research was supported by the State Key Laboratory of Road Engineering Safety and Health in Cold and High-altitude Regions (No. YGY2017KYPT-04), the National Natural Science Foundation of China (No. 41971076), and the National Natural Science Foundation of China (No. 42171128).

**Institutional Review Board Statement:** Not applicable.

**Informed Consent Statement:** Not applicable.

**Data Availability Statement:** Not applicable.

**Conflicts of Interest:** The authors declare no conflict of interest.

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
