# Peer review of "Settlement Characteristic of Warm Permafrost Embankment with Two-Phase Closed Thermosyphons in Daxing’anling Mountains Region"

_sustainability, doi:10.3390/su141912272_

Round 1
Reviewer 1 Report
This manuscript performed a field test and numerical calculation to investigate the the thermal status and the deformation characteristics of an embankment with TPCTs, and discussed the cooling effects of locations, depths, and shapes of TPCTs. The manuscript is well-organized and of important practical values for TPCTs in the permafrost regions of Da-xing'anling Mountains. The reviewer recommends accepting the paper after the authors properly addressing the following questions:
(1) For the Introduction part, recommend the authors to include more latest studies on the effect of the TPCT embankment.
(2) Please check the format and spelling of the article, e.g, Line 106 and 166, modify the paragraph format, Line 151 and 198, “is” change to “are”.
(3) Line 131: “TS” and “TL” have been explained in the article, the authors need to delete the repetitive instructions. Line 130, “Tc” was set as -1.5℃, Line167, “n” is generally 0.6 in this paper. The authors need to explain how to determine the value of the parameter, based on derivation or cited from other papers.
(4) Figure 6: the start and end time of field test data of this figure need to be indicated. In Figure 9, the legend in the figure needs to be aligned.
(5) 3.1: temperature field in 50 years section, recommend the authors add the calculation method of initial foundation temperature in the numerical models.
(6) 4.1: the authors need to add the description of geometric parameters for the TPCTs.
(7) Line 465-468: need to include the influence of TPCTs at different locations on the cooling effect of the embankment.
Reviewer 2 Report
The research topic is important and interesting, and lots of work has been done in the manuscript. The authors were suggested to further revise some issues. Some suggestions:
1. Figure 1 was not well presented. It’s hard to distinguish different elements.
2. The working principle of TPCT has not clearly stated in the section of Introduction. It is recommended to use a schematic diagram to represent its structure, working method, etc.
3. In the section of Introduction, the author emphasized that asphalt pavement structures are mainly used in permafrost regions. But how did this paper choose the cement pavement structure as the analysis object according to the Figure 2? And the contents shown in the left and right pictures of Figure 2 did not correspond exactly.
4. A lot of formulas were used in the text, please double check the correctness of all formulas.
5. Figure 3 was incomplete and did not show the on-site monitoring process.
6. According to the Figure 5, it is not appropriate to use only one set of data to prove that the calculated and measured values are in good agreement.
7. “Mpa” should be “MPa”.
8. According to the statement according to the Figure 5, the calculated value and measured value were in good agreement. But why did Figure 7 show that the two results were different?
9. When using the finite element models to simulate the settlement of highway within 50 years after the construction, the choice of simulation parameters is very important. How is the rationality of parameters considered in this article, please highlight the content of this part.
Reviewer 3 Report
This study focused on the long-term temperature and deformation field of the permafrost under the Mohe-Beijicun Highway. Through the finite element method, the changes of permafrost embankment with and without two-phase closed thermosyphons (TPCTs) in 50 years were studied. The contribution of different TPCTs to the cooling of permafrost and reducing embankment settlement was assessed. Results show that the TPCTs have an excellent cooling effect on the permafrost embankment. In order to ensure the cooling effect and reduce the differential settlement of the embankment, it is suggested that L-shaped TPCTs should be used in remedial engineering.
1. It is suggested to axis headings for supplementary figure (Figure 10).
2. The use of vertically installed TPCTs has a better cooling effect on permafrost. L-shaped TPCTs have the best effect on reducing differential settlement. It is suggested to supplement the basis for balancing the link between the two factors.
3. It is suggested to supplement the difference between flexible L-type TPCT and traditional L-type TPCT in section 4.3.
4. For L-type TPCT, compared with other types of TPCT, supplementary quantitative analysis is more convincing in improving the effect of uneven settlement.
Reviewer 4 Report
Before making any recommendations for an interesting scientific article "Thermal and Settlement Characteristics of Warm Permafrost Embankment with Two-phase Closed Thermosyphons in Daxing'anling Mountains Region", I would like to present the following statements on the topic. Based on my long-term research and transfer profile in the field of holistic perception of the issues of pavement engineering, I consider the evaluated scientific article to be extremely topical and fully convergent with the following author's research and educational premise. Pavements should be designed, built, managed, maintained, recycled (decomposed) at a reasonable price, in reasonable quality, respecting the relevant requirements of users, residents and sustainable development, in this case also in the most climatically demanding locations.
Mandatory requirements:
LNSA 6 (Line Number of Scientific Article)...what does CCCC mean for?
LNSA 28-90. 1. Introduction... Climate pavement adaptability is an integral part of a holistic concept of road design, construction, and pavement management in the all World. The Introduction needs to be expanded with global knowledge from other continents as well.
LNSA (Line Number of Scientific Article) 29-30... a large number of asphalt concrete pavement and less concrete pavement highway were constructed on it,.. I recommend using the plural number and the term only asphalt pavements.
LNSA 159... elastic modulus E (Mpa)...it is necessary to remove the typo in the marked MPa.
LNSA 269... Figure 10. Embankment settlement from 2010 to 2060 (cm)…it would be appropriate to explicitly state the descriptions of the x and y axes on the individual modeled embankment settlement courses.
LNSA 285... Figure 11. Change of settlement in different locations... the marking of the x-axis needs to be corrected.
LNSA 443-464...5. Conclusions.. considering the amount of work done and its quality, it would be appropriate to expand the conclusions. It is necessary to highlight the authors' own contribution and the achieved results to the set goals (introduction extension required). As already indicated, the results of the authors' research need to be compared with important articles by foreign authors, including Europe.
LNSA 443-464...5. Conclusions.. considering the amount of work done and its quality, it would be appropriate to expand the conclusions. It is necessary to highlight the authors' own contribution and the achieved results to the set goals (introduction extension required). As already indicated, the results of the authors' research need to be compared with important articles by foreign authors, including Europe.
Facultative recommendations:
LNSA 2-4… I consider the paper name "Thermal and Settlement Characteristics of Warm Permafrost Embankment with Two-phase Closed Thermosyphons in Daxing'anling Mountains Region" too long. In the context of the presented facts, I would choose for example: (Settlement of Permafrost Embankment with Two-phase Closed Thermosyphons in Daxing'anling Mountains Region,...)
LNSA 157...Table 1. Thermal parameters of each layer... Were the presented values of Layer Heat conductivity coefficient for cement concrete obtained by own measurements? What are the stated values in relation to the standard (design) values of the characteristics used in structural design of pavements? In Central European conditions for the prevail design value 2.5 (W.m-1 .K-1).
LNSA 233-236… During the 40 years from 2020 to 2060, the temperature of embankment and foundation is gradually increasing, and the increased range of embankment is significantly greater than that of foundation due to the increase of annual average temperature outside… This finding it seems too bold to me, it would be appropriate present under what assumptions and with what degree of reliability itcould be expected.
LNSA 280-281... In the first 25 years, the settlement rate is about 0.9 cm/year, and in the second 25 years, it is about 1.5 cm/year... I consider the time period of 25 years to be too long in case of the significantly non-linear course of embankment settlement.
LNSA 355... Figure 15. Lateral distribution of road surface settlement after 1 year with different depth of TPCTs... how can you explain the significantly different course of the function for 8 m than as is in the cases depths of 4 and 6 m?
LNSA 469-429...References…from the aspect of the fact that the investigated problem is a problem of many countries, I consider the number of references 26 to be insufficient. As already indicated, it is necessary to add references from other continents, including Europe.
The reviewed scientific article as very good, containing certain shortcomings that need to be eliminated. From the aspect of quality of reviewed article, in case of incorporation of comments, or relevant justification of their non-incorporation, I am able to process a repeated review within 3 days.
Round 2
Reviewer 2 Report
The manuscript has been well revised.
Author Response
Thank you very much for your approval of our manuscript revision.
Reviewer 4 Report
By the author of the article " Settlement Characteristic of Warm Permafrost Embankment with Two-phase Closed Thermosyphons in Daxing'anling Mountains Region" (original title Thermal and Settlement Characteristics of Warm Permafrost Embankment with Two-phase Closed Thermosyphons in Daxing'anling Mountains Region), I would like to present this summary evaluation. I am very pleased to be able to sincerely congratulate the authors on the excellent quality of the second version of their scientific contribution. The authors also took great care to incorporate my mandatory and facultative recommendations, although they certainly took a considerable amount of time. I sincerely thank the authors for the opportunity to expand my territorially limited knowledge about permafrost and familiarize themselves with the new findings presented in the second version of the article under consideration. I keep my fingers crossed for publishers in sustainable improvement of their very useful magazine, for my scientific profilation of holistic perception of sustainable pavement design, construction, management and recycling. Based on my long-term research and transfer profile in the field of holistic perception of the issues of pavements , I consider the evaluated scientific article to be extremely topical and fully convergent with the following author's research and educational premise. Pavements should be designed, built, managed, maintained, recycled (decomposed) at a reasonable price, in reasonable quality, respecting the relevant requirements of users, residents in their surroundings and sustainable development principles.
I would allow recommend to the authors to consider incorporating the following recommendations or requirements.
LNSA (Line Number of Scientific Article) 157-159 ... Figure 2. Geological conditions and geometric model: (a) Diagram of geological conditions, (b) Geometric model and mesh division... it would speed up the reader's understanding of the figure if part of the legend related to the structural composition roadway, was moved above the composition of its subsoil (gravel soil,...).
LNSA 168…equations 13 and 15 are identical.
LNSA 175... Table 1. Thermal parameters of each layer... I recommend unifying the format of writing physical units.
LNSA 185...TPCT..when using the abbreviation TPCT for the first time in the title of the chapter, I recommend that you also state its meaning.
LNSA 231... Figure 6. Ground temperature curve (a) measured value; (b) calculated value...I recommend using the plural number (curves, values) in the name of the figure, it also applies to figures 7, 8.
LNSA 274.. Figure 9. Variation curve of embankment centerline temperature in different layers with time...I recommend reformatting the texts so that the figure and its title are on one side, also applies into figure 17.
LNSA 329-332... Figure 12. The temperature change of embankment centerline with different locations of TPCTs: (a) time-history curve of embankment temperature (b) temperature of embankment after 10 years... I recommend adding a graphic identification into the figure 12(shoulder, slope toe, berm,...) or add these names to figure 16.
LNSA 395-396...Figure 16. Different shapes of TPCTs: (a) vertically installed TPCT, (b) obliquely installed TPCT, (c) flexible L-shaped TPCT...I recommend that the authors consider adding at least indicative dimensions to the figure.
LNSA 584...a period needs to be added at the end of reference 33, it also applies to 36.
In conclusion, I would like to abstract the presented facts into repeated thank and congratulations to the authors as well as the publisher of inspirational scientific journal Sustainability.
